# Deceptive Opponent Modeling with Proactive Network Interdiction for Stochastic Goal Recognition Control

## Abstract

Goal recognition based on the observations of the behaviors collected online has been used to model some potential applications. Newly formulated problem of goal recognition design aims at facilitating the online goal recognition process by performing offline redesign of the underlying environment with hard action removal. In this paper, we propose the stochastic goal recognition control (S-GRC) problem with two main stages: (1) deceptive opponent modeling based on maximum entropy regularized Markov decision processes (MDPs) and (2) goal recognition control under proactively static interdiction. For the purpose of evaluation, we propose to use the worst case distinctiveness ($wcd$) as a measure of the non-distinctive path without revealing the true goals, the task of S-GRC is to interdict a set of actions that improve or reduce the $wcd$. We empirically demonstrate that our proposed approach control the goal recognition process based on opponent's deceptive behavior.

## 1 Introduction

Goal recognition (GR), also called intention recognition, is the task of inferring the goals of an agent according to the observed actions or states collected online (Sadri, 2011), which enables humans or agents to make proactive response plans. Goal recognition design (GRD) is to deliberately redesign the environment for improved online goal recognition ability (Keren et al., 2014), which includes two key models, the first measures how efficiently and effectively the online goal recognition system performs in a given setting, the second optimizes the goal recognition setting via redesign (Keren et al., 2019). Goal recognition control (GRC) aims at soft action interdiction under bounded resource by adding cost compared to the hard action removal of GRD (Luo et al., 2019), in which the interdiction can be proactively static inhibition or online dynamic block by allocating security resources to protect goals against the attacker.

In fact, under complex environment (both adversarial and cooperative), the information of an opponent's goals are asymmetric and can not be obtained through communication (Le Guillarme, 2016). Most recent advancements in GR utilize planning recognition as planning (PRAP) (Ramírez & Geffner, 2009), generative game-theoretic frameworks (Ang et al., 2017), and goal recognition as planning (GRAP) (Pereira et al., 2019). There are three key assumptions widely used: (1) agents performs optimal plans to real the goals; (2) the environment is fully observable, that is the states and actions of the agents are observable; and (3) the agent's actions are deterministic. However, these existing frameworks seldom address the deceptive behaviors of actively misleading the goal recognition process. We will relax the first and third assumptions to handle non-optimal agents and stochastic actions. Such as one game-theoretic approach provided with a unified treatment of both threat assessment and response planning in (Guillarme et al., 2017), after evaluating the threat, the defender will allocate road barrier or patrolling force to protect critical infrastructure, as illustrated in Figure 1.

In order to identify the goals and improve security, we employ opponent modeling to deal with the non-stationary strategies stemming from deception, in which maximum entropy regularized Markov decision process (MDP) is utilized to shape the multi-modal adversarial strategies (Shen & How,

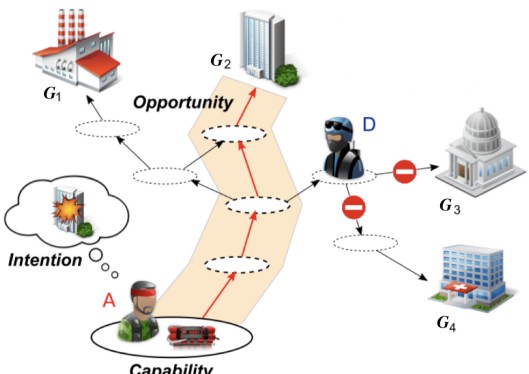

Figure 1: Illustration of critical infrastructure protection (CIP) (Guillarme et al., 2017): the defender can perform proactive road network interdiction to accelerate the opponent's goal recognition even with possible deceptive behaviors.

2019). After finding the maximal non-distinctive agent path, we are seeking some optimal modifications (e.g., interdiction to block or inhibit action) to accelerate the online goal recognition.

In this work, we propose the stochastic goal recognition design (S-GRC) problem with deceptive opponent modeling and proactive network interdiction: the opponent's deceptive policy is modeled as one soft multi-criteria decision policy with one tunable parameter to balance goal achievement and deception preference; and the primary objective is offline redesigning the environment by soft action interdiction to control online GR.

To validate the model, we evaluate our approach in two different environment representations: random generated connected graph and real road network. Our experiments demonstrate that (1) deceptive opponent modeling with entropy regularization make it robust to multi-modal stochastic behaviors; (2) soft decision policy based stochastic goal recognition design model for action removal bridge the gap between observation and decision making.

## 2 RELATED WORK

### 2.1 GOAL RECOGNITION AND DESIGN

**Goal Recognition (GR):** The ability of inferring the goals of others can assist us to reason about what they are doing, why they are doing it, and what they will do next (Sukthankar et al., 2014). The recognition setting can be generally be divided into keyhole, where the agent is unaware of being observed and recognized as if the agent is looking through a keyhole; adversarial, where the agent is actively hostile to the observations of the actions and the inference of the goals; and intended, where the agent wants to convey the goal to be understood and helps the recognizer to detect the objective (Keren et al., 2019). In this work, we focus on the adversarial setting, where the agent would hide the goal and attempt to thwart the recognition process by deception. GR is strongly related to the topic of privacy-preserving and explainable planning (Keren et al., 2016; Chakraborti et al., 2019; Kulkarni et al.).

**Goal Recognition Design (GRD):** The GRD problems contains two models: one goal recognition setting analyzed environments, acting agent (attacker), and recognition system (recognizer, observer); one design model of possible ways to change the environment (Keren et al., 2019). The environment will induce a set of possible behaviors for the attacker, GRD can be divided into deterministic GRD (Keren et al., 2019), stochastic GRD (S-GRD) (Wayllace et al., 2018; Wayllace, 2019), GRD for Agents with partial knowledge (GRD-APK) (Sarah et al., 2019), and game-theoretic GRD (Ang et al., 2017; Masters & Sardina, 2017). Also, many design measures have been employed, such as action removal (AR), action sensor refinement (SAR), and action conditioning (AC) (Keren et al., 2018; Wayllace et al., 2018).

## 2.2 OPPONENT MODELING AND NETWORK INTERDICTION

**Deceptive Opponent Modeling:** Opponent modeling has been employed to reason the non-stationary strategies of autonomous agents (Albrecht & Stone, 2018; Papoudakis et al., 2019). Deception involves combination of hiding the truth and showing the false (Masters & Sardina, 2017), which articulated three distinct strategies in each case: dissimulation (masking, repackaging, and dazzling), simulation (mimicking, inventing, and decoying) (Ettinger & Jehiel, 2010). As for deceptive path planning, game-theoretic method (Root, 2005) and model based (Masters & Sardina, 2017) method have been investigated.

**Network Interdiction:** In recent researches, interdiction has been used to counter the attacker's behavior, MDP interdiction (Panda & Vorobeychik, 2017), plan interdiction (Vorobeychik & Pritchard, 2018), and network interdiction (Xu et al.) show one promising research thread of Observe-Orient-Decide-Act (OODA).

## 2.3 REGULARIZED MDPS

Most sequential decision-making problems can be described as MDP based reinforcement learning problems or stochastic game problems. Regularization for MDPs can be divided into temporal regularization problems in temporal space (Thodoroff et al., 2018), spatial regularization in feature or state space (Farahmand, 2011; Harrigan), and entropy or information regularization in policy space (Neu et al., 2017; Geist et al., 2019; Belousov & Peters, 2019). Regularized reinforcement learning focuses on the smoothness and sparsity of the value function (Farahmand, 2011), acquiring of diverse robot skills (Haarnoja, 2018), and sparsity and multi-modality of the optimal policy (Lee et al., 2019). Regularized stochastic game employees various entropy constraints to balance or control behavior (Grau-Moya et al., 2018; Savas et al., 2019; Ahmadi et al., 2018; Tian et al., 2019).

## 3 STOCHASTIC GOAL RECOGNITION CONTROL

The goal recognition control (GRC) problems try to balance the intended and adversarial goal recognition process under network interdiction (Luo et al., 2019), which extends to assume that the agent can choose to reveal or obfuscate (Kulkarni et al., 2019b;a), share or hide (Strouse et al., 2018) the goals. In this paper, we focus on stochastic GRC (S-GRC) under adversarial, cooperative, and non-stationary environment. The framework of our proposed approach is illustrated in Figure 3.

We embed deceptive opponent modeling and proactive network interdiction into the OODA cycle as two modules, which could bridge the gap between observation and decision-making. The deceptive opponent modeling module will shape the multi-modal opponent, the proactive network interdiction module will provide bounded resource allocation for defense. So, GRC will identify the goals with topological soft q learning and provide proactive interdiction resource allocation.

## 3.1 SSP-MDP FOR S-GRC

Different from the GRD problem, we employ more applicable and soft measure of interdiction with bounded resource to control the goal recognition process. With this motivation in mind, we use the Stochastic Shortest-Path MDP (SSP-MDP) (Kolobov & Kolobov, 2012) model, which is widely used for uncertainty given an initial state, a set of goal states, actions with probabilistic outcomes and an action cost function. A SSP-MDP is represented as a tuple $\langle \mathbf{S}, s_0, \mathbf{A}, \mathbf{T}, \mathbf{C}, \mathbf{G} \rangle$, which consists a set of states $\mathbf{S}$, a initial state $s_0$, a set of actions $\mathbf{A}$, a transition function $\mathbf{T} : \mathbf{S} \times \mathbf{A} \times \mathbf{S} \to [0, 1]$ of state transition from $s$ to $s'$ after executing the action $a$, and a cost function $\mathbf{C} : \mathbf{S} \times \mathbf{A} \times \mathbf{S} \to \mathbb{R}$ that gives the cost of executing action $a$, and a set of terminal goals $\mathbf{G} \subseteq \mathbf{S}$. With a cost function to replace the reward function, the objective is to find a optimal policy $\pi$ so as to minimize the smallest expected cost or travel time of the policy, which forms the map from states to actions.

So, the optimal policy of the SSP-MDP can be find by using the Value Iteration (VI) algorithm, which use a value function $V$ to represent expected costs. The expected cost of an optimal policy $\pi^*$ for the starting state $s_0 \in \mathbf{S}$ is the expected cost $V(s_0)$, and the expected cost for all states $s \in \mathbf{S}$ can be calculated by the Bellman equation (Kolobov & Kolobov, 2012):

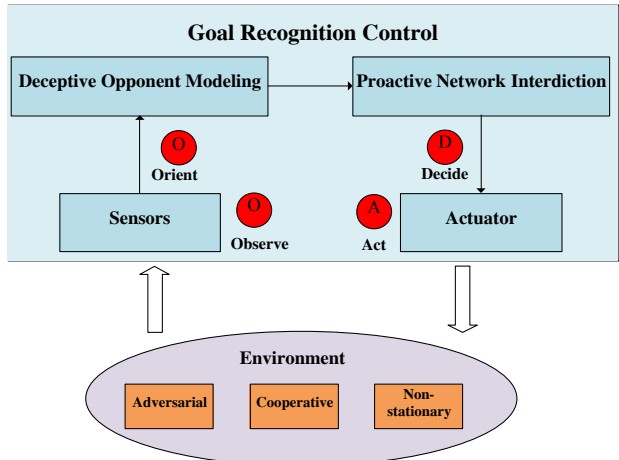

Figure 2: The OODA cycle embedded with stochastic goal recognition control (S-GRC): one deceptive opponent modeling module and one goal recognition control module.

$$V(s) = \min_{a \in A} \sum_{s' \in \mathbf{S}} T(s, a, s') \left[ C(s, a, s') + V(s') \right] \tag{1}$$

A previous work on stochastic shortest path network interdiction modeled the problem as one sequential bi-level attacker-defender game, where the defender moves first by deploying sensors to the arcs in the network maximizing the expected shortest traveling time of the adversary (Zhang et al., 2018). Here we formulate the stochastic goal recognition control (S-GRC) problem as an extension of the GRD problem to allows non-optimal agents and stochastic actions, which means the agent's action are not deterministic and the possible successor states are with probability, the measures we can take are interdictions with cost. The bounded interdiction resources allocations can better model the real world applications. A S-GRC problem is represented as a tuple $P = \langle D, \mathbf{G} \rangle$, the domain information $D = \langle \mathbf{S}, s_0, \mathbf{A}, \mathbf{T}, \mathbf{C}, \mathbf{G} \rangle$ can be represented as SSP-MDPs. Similar to the S-GRD (Wayllace et al., 2018), we employ a metric to assess the largest number of actions an agent can take or the largest cost an agent will incur before revealing the goal. Given a problem $P = \langle D, \mathbf{G} \rangle$, the worst case distinctiveness ($wcd$) of the problem can be defined as:

$$wcd(P) = \max_{\pi \in \mathbf{\Pi}} V_\pi(s_0) \tag{2}$$

So the objective of S-GRC under network interdiction can be reformulated as one optimization problem of finding a subset of actions $\Delta \mathbf{A}^* \subset \mathbf{A}$:

$$
\begin{aligned}
\Delta \mathbf{A}^* = \underset{\Delta \mathbf{A} \subset \mathbf{A}}{\operatorname{argmin}} \, wcd(\hat{P}) \\
\text{s.t.} \quad V_{\pi;s}(s_0) = V_{\hat{\tau};\hat{s}}(s_0) \quad \forall g \in \mathbf{G} \\
\sum (\mathbb{1}_{\Delta \mathbf{A}^*} \times \mathbf{I}) \le k
\end{aligned}
\tag{3}
$$

where $P = \langle \hat{D}, G \rangle$ is the problem with domain $\hat{D} = \left\langle \mathbf{S}, s_0, \hat{\mathbf{A}}, \mathbf{T}, \mathbf{C}, \mathbf{G} \right\rangle$, $\hat{\mathbf{A}}$ is the redesigned action set with $\Delta \mathbf{A}^*$ interdicted, $\mathbf{I}$ is the corresponding cost to the interdiction resource, the total interdiction resource allocation is bounded by $k$.

### 3.2 Augmented SSP-MDP with SCCs

With the metric $wcd$, we can measure the maximal number of actions an agent can take before its goal is revealed. However, the formal $wcd$ related metrics for S-GRDs are inconsistent with the intuitive definition, the $wcd$ computation is not Markovian owing to the set of possible goals depend on the observed trajectory to a state. To deal with the uncertain inconsistence, $wcd_{ag}$ and

$ECD$ together with topological value iteration are employed to model the problem, in which the augmented MDPs with strongly connected components do not have loops (Wayllace et al.).

The state space of an MDP can be represented as a directed connectivity graph. One example MDP is shown in Figure 3(a), where the states are denoted by nodes, the actions are denoted by edges, the transitions are related to the arrow. The $wcd$ may be infinite when the state space contains some cycle loop, here we can use the Tarjan algorithm (Tarjan, 1972; Hou et al., 2014) to generate strongly connected components (SCCs), which form a directed acyclic graph (DAG) as shown in Figure 3(b).

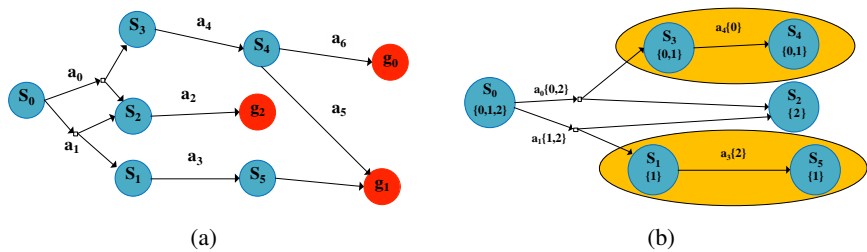

Figure 3: Augmented MDP with strong connected components (SCCs).

**Augmented SSP-MDP for S-GRC**: Given a SSP-MDP, an augmented SSP-MDP $\left\langle \tilde{\mathbf{S}}, \tilde{s}_0, \tilde{\mathbf{A}}, \tilde{\mathbf{T}}, \tilde{\mathbf{C}}, \tilde{\mathbf{G}} \right\rangle$ augments each component of the tuple:

- Each state $\hat{s} \in \hat{\mathbf{S}}$ is represented by $\langle s, \mathbf{G}' \rangle$.

- The augmented initial state is $\tilde{s}_0 = \langle s_0, \mathbf{G} \rangle$.

- Each augmented action $\hat{a} \in \hat{\mathbf{A}}$ is a tuple $\langle a, \mathbf{G}' \rangle$, where $a\mathbf{A}$ and $\mathbf{G}'$ is the set of all goals for which that action is an optimal action.

- The new transition function $\tilde{\mathbf{T}} : \tilde{\mathbf{S}} \times \tilde{\mathbf{A}} \times \tilde{\mathbf{S}} \rightarrow [0,1]$ gives the probability $\tilde{T}(\tilde{s}, \tilde{a}, \tilde{s}')$, where $\tilde{s} = \langle s, \mathbf{G}' \rangle$, $\tilde{a} = \langle a, \mathbf{G}' \rangle$, and $\tilde{s}' = \langle s', \mathbf{G}' \cap \mathbf{G}'' \rangle$, $\tilde{T}(\tilde{s}, \tilde{a}, \tilde{s}') = T(s, a, s')$, if $|\mathbf{G}' \cap \mathbf{G}''| > 1$ and equals 0 otherwise.

- The cost function $\tilde{C} : \tilde{S} \times \tilde{A} \times \tilde{S} \rightarrow \mathbb{R}^+$ gives the cost $\tilde{C}(\tilde{s}, \tilde{a}, \tilde{s}')$ of executing action $\tilde{a}$ in augmented state $\tilde{s}$ and arriving in $\tilde{s}\prime\prime$. This cost equals the cost $\tilde{C}(\tilde{s}, \tilde{a}, \tilde{s}') = C(s, a, s')$ under the same condition as above.

- The augmented goal states $\tilde{\mathbf{G}} \subseteq \tilde{\mathbf{S}}$ are those augmented states $\langle s, \mathbf{G}' \rangle$ for which any execution of an augmented action will transition to an augmented state $\langle s', \mathbf{G}''' \rangle$ with one goal or no goals.

### 3.3 DECEPTIVE OPPONENT MODELING

As the opponent may change the goals midway to mislead the goal recognition process. We use a binary variable $\lambda \in \{0, 1\}$ to characterize whether the opponent is a optimal or deceptive with decoy goals.

Depending on $\lambda$, the opponent is expected to exhibit different behaviors, which is fully described by an opponent policy $\pi^o(a_t^o|s_t)$. This model is restrictive since the action probability only depend on the current state. Nonetheless, we use this model only for policy learning, and use a general history dependent opponent policy for the evaluation of the learned autonomous agent policy. Another implicit assumption of this model is that the opponent has full observability over the states. This assumption could be released by modeling the opponent as a POMDP agent.

Neutral Opponent: If the opponent is a civilian, i.e. $\lambda = 0$, we assume a simple reactive policy $\pi^{\text{cil}}(a_t^o|s_t)$ is available to model the opponent:

$$\pi^o(a_t^o|s_t, \lambda = 0) = \pi^{cil}(a_t^o|s_t). \tag{4}$$

Table 1: Sample table title

| PART | DESCRIPTION |
|---|---|
| Dendrite | Input terminal |
| Axon | Output terminal |
| Soma | Cell body (contains cell nucleus) |

Deceptive Opponent: We use the following equation to model an adversarial agent's policy $\pi^o$:

$$\pi^o(a_t^o|s_t, \lambda = 1; \alpha, \beta) = \text{argmin}_{\pi \in \Delta}\{\mathbb{KL}(\pi|\pi_\alpha^{\text{MDP}}) + \beta\mathbb{KL}(\pi|\pi^o(\cdot|s_t, \lambda = 0))\}, \tag{5}$$

$$\pi_\alpha^{\text{MDP}}(a_t^o|s_t, \lambda = 1) = e^{\alpha Q(s_t, a_t^o)}/Z(s_t), \tag{6}$$

where $\mathbb{KL}(\cdot|\cdot)$ denotes the Kullback–Leibler divergence between two distributions, The goal-achieving policy $\pi_\alpha^{\text{MDP}}$ is associated with the optimal Q function $Q(s_t, a_t^o)$, of a goal-achieving adversary MDP defined later. The temperature parameter $\alpha$ in (6) represents the level of rationality of the adversary. The other parameter $\beta$ indicates the level of deception. $Z(s_t)$ is the partition function that normalizes $\pi_\alpha^{\text{MDP}}$.

**Soft Q Learning**: We use soft-Q learning (Haarnoja, 2018) to learn a stochastic belief space policy. The soft Q learning objective is to maximize the expected reward regularized by the entropy of the policy,

$$\sum_t \mathbb{E}_{b_t, s_t, a_t \sim \rho_\pi} \gamma^t[r(b_t, s_t, a_t) + \sigma H(\pi(\cdot|b_t, s_t))]. \tag{7}$$

The parameter $\sigma$ controls the 'softness' of the policy. The nice interpretation of this objective function is maximizing accumulative reward while behaving as uncertain as possible, which is a desired property against an adversary.

This maximum entropy problem is solved using soft Q iteration. For discrete action space, the fixed point iteration:

$$Q_{\text{soft}}(b_t, s_t, a_t) \leftarrow r_t + \gamma\mathbb{E}_{b_{t+1}, s_{t+1} \sim p_s}[V_{\text{soft}}(b_{t+1}, s_{t+1})], \tag{8}$$

$$V_{\text{soft}}(b_t, s_t) \leftarrow \sigma \log \sum_{a \in A} \exp(\frac{1}{\sigma}Q_{\text{soft}}(b_t, s_t, a)), \tag{9}$$

converges to the optimal soft value functions $Q_{\text{soft}}^*$ and $V_{\text{soft}}^*$ (Haarnoja, 2018), and the optimal policy can be obtained from:

$$\pi_{\text{MaxEnt}}^*(a_t|b_t, s_t) = \exp(\frac{1}{\sigma}(Q_{\text{soft}}^*(b_t, s_t, a_t) - V_{\text{soft}}^*(b_t, s_t))). \tag{10}$$

### 3.4 PROACTIVE NETWORK INTERDICTION

Proactive network interdiction

## 4 EXPERIMENTS

### 4.1 GRID WORLD

### 4.2 NETWORK GRAPH

### 4.3 RESULTS

## 5 CONCLUSION AND FUTURE WORK

Risk-Sensitive MDPs (Hou et al., 2014) illustrate one applicable scenario where the cost function is constrained with one threshold or deadline.

See Table 1.

ACKNOWLEDGMENTS

This work is partially sponsored by the National Natural Science Foundation of China under Grants No. 61702528, No. 61806212.

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
