# OpenReview forum: "Deceptive Opponent Modeling with Proactive Network Interdiction for Stochastic Goal Recognition Control"
_ICLR.cc/2020/Conference — Reject_

### Official Review · AnonReviewer2 · 2019-10-12
**Official Blind Review #2**

**Rating:** 1

**Review:**

This paper studies goal recognition control given a deceptive opponent, who selects actions to intentionally mislead or confusing the learner to learn the true goal. The problem has been studied in the security game, resource allocation game and Stackelberg game, where the defender is to play a resource allocation game given the best response of the attacker. in this paper, the authors use stochastic-shortest-path MDP to model the attacker's planning problem. The defender's objective is stated in eq (2) and (3) but not explained clearly.

The work considers a deceptive attacker whose objective is not to maximize the total reward/minimize the total cost, but rather to balance two objectives: minimizing the total cost while maximizing the KL divergence between its current policy and the optimal goal-achieving policy. This opponent model seems to be simplistic, given security game has studied the opponent using more rigorous equilibrium analysis and with dynamic Bayesian game formulation.

see Manshaei, Mohammad Hossein, et al. "Game theory meets network security and privacy." ACM Computing Surveys (CSUR) 45.3 (2013): 25.

The practical application is well motivated.  The paper itself is incomplete and contains many typos. The paper is unfinished.

**Experience Assessment:**

I have read many papers in this area.

**Review Assessment: Checking Correctness Of Derivations And Theory:**

N/A

**Review Assessment: Checking Correctness Of Experiments:**

N/A

**Review Assessment: Thoroughness In Paper Reading:**

I read the paper thoroughly.

---

### Official Review · AnonReviewer1 · 2019-10-22
**Official Blind Review #1**

**Rating:** 1

**Review:**

This paper proposes to use reinforcement learning to model an agent that is reaching goal that it is intended to reach. The authors consider the case where the agent is (1) indifferent to being observed, (2) trying to help an observer reach its goal, and (3) trying to fool the observer into not reach its goal. The paper propose to use a value function to quantify how easy it is to predict where the agent is going ("worst case distinctiveness"). The authors propose to then train an agent to modify the action space to make it difficult for an agent to fool the observer.

While this is an interesting idea, the paper is clearly incomplete. The experiment section is missing, and much of the method section is partly written.

**Experience Assessment:**

I have published one or two papers in this area.

**Review Assessment: Checking Correctness Of Derivations And Theory:**

I assessed the sensibility of the derivations and theory.

**Review Assessment: Checking Correctness Of Experiments:**

N/A

**Review Assessment: Thoroughness In Paper Reading:**

I read the paper at least twice and used my best judgement in assessing the paper.

---

### Official Review · AnonReviewer4 · 2019-10-30
**Official Blind Review #4**

**Rating:** 1

**Review:**

This paper aims to provide a goal recognition framework. The paper reviews previous literature and outlines a model, but it seems to be at the draft stage as it stands as the experiment section is missing (section 4). Also, in some parts the sentences are broken and hard to follow, for example, “Here we formulate the stochastic goal recognition control (S-GRC) problem as an extension of the GRD problem to allows non-optimal agents and stochastic actions, which means the agent’s action are not deterministic and the possible successor states are with probability, the measures we can take are interdictions with cost.”
Based on the above comments I think the paper is not ready for publication.


**Experience Assessment:**

I do not know much about this area.

**Review Assessment: Checking Correctness Of Derivations And Theory:**

I did not assess the derivations or theory.

**Review Assessment: Checking Correctness Of Experiments:**

N/A

**Review Assessment: Thoroughness In Paper Reading:**

I made a quick assessment of this paper.

---

### Public Comment · ~Junren_Luo1 · 2019-11-08
**Submission Withdrawn by the Authors**

I have read and agree with the venue's withdrawal policy on behalf of myself and my co-authors.

---

### Decision · Program_Chairs · 2019-12-19

**Decision:**

Reject

**Comment:**

This paper has been withdrawn by the authors.